

# Wood microclimate as a predictor of carbon dioxide fluxes from deadwood in tropical Australia

Elizabeth S. Duan[1,2,*], Luciana Chavez Rodriguez[2,*], Nicole Hemming-Schroeder[3], Baptiste Wijas[4,5], Habacuc Flores-Moreno[6], Alexander W. Cheesman[7], Lucas A. Cernusak[7], Michael J. Liddell[7,8], Paul Eggleton[9], Amy E. Zanne[5], and Steven D. Allison[2,3]

[1]Department of Biology, University of Washington, Seattle, Washington, USA
[2]Department of Ecology and Evolutionary Biology, University of California Irvine, Irvine, California, USA
[3]Department of Earth System Science, University of California Irvine, Irvine, California, USA
[4]School of Biological, Earth and Environmental Sciences, University of New South Wales, Sydney, NSW, Australia
[5]Department of Biology, University of Miami, Miami, FL, USA
[6]Commonwealth Scientific and Industrial Research Organisation, Brisbane, QLD, Australia
[7]College of Science and Engineering, James Cook University, Cairns, QLD, Australia
[8]Centre for Tropical Environmental and Sustainability Science, James Cook University, Cairns, QLD, Australia
[9]The Soil Biodiversity Group, Entomology Department, The Natural History Museum, London, UK
[*]These authors contributed equally to this work.

**Correspondence:** Luciana Chavez Rodriguez (lucianac@uci.edu)

**Abstract.**

Deadwood is an important yet understudied carbon pool in tropical ecosystems. Wood microclimate, as defined by wood moisture content and temperature, drives decomposer (microbial, termite) activities and deadwood degradation to $CO_2$. Microclimate is strongly influenced by local climate, and thus, climate data could be used to predict $CO_2$ fluxes from decaying wood. Given the increasing availability of gridded climate data, this link would allow the rapid estimation of deadwood-related $CO_2$ fluxes from tropical ecosystems worldwide. In this study, we adapted a mechanistic fuel moisture model that uses weather variables (e.g. air temperature, precipitation, solar radiation) to characterize wood microclimate along a rainfall gradient in Queensland, Australia. We then developed a Bayesian statistical relationship between microclimate and $CO_2$ flux from pine (*Pinus radiata*) blocks deployed at sites and combined this relationship with our microclimate simulations to predict $CO_2$ fluxes from deadwood at 1-hour temporal resolution. We compared our pine-based simulations to moisture-$CO_2$ relationships from stems of native tree species deployed at the wettest and driest sites. Finally, we integrated fluxes over time to estimate the amount of carbon entering the atmosphere and compared these estimates to measured mass loss in pines and native stems. Our statistical model showed a positive relationship between $CO_2$ fluxes and wood microclimate variables. Comparing cumulative $CO_2$ with wood mass loss, we observed that carbon from deadwood decomposition is mainly released as $CO_2$ regardless of the precipitation regime. At the dry savanna, only about 19% of the wood mass loss was released to $CO_2$ within 48 months, compared to 86% at the wet rainforest, suggesting longer residence times of deadwood compared to wetter sites. However, the amount of carbon released in-situ as $CO_2$ is lower when wood blocks are attacked by termites, especially at drier sites. These results highlight the important but understudied role of termites in the breakdown of deadwood in dry climates. Additionally, mass loss-flux relationships of decaying native stems deviated from that of pine blocks. Our results indicate that wood micro-



climate variables are important in predicting $CO_2$ fluxes from deadwood degradation, but are not sufficient, as other factors such as wood traits (wood quality, chemical composition, and stoichiometry) and biotic processes should be considered in future modeling efforts.

## 1   Introduction

Tropical and subtropical forests are important ecosystems in the global terrestrial carbon (C) cycle (Raich et al., 2006; Mitchard,
2018; Taylor et al., 2017). In 2020, they made up 61% of the global tree cover by area (FAO, 2020). Within tropical forests, deadwood, including fallen trees and branches, stumps, and dead standing trees (Woldendorp and Keenan, 2005), can account for more than 50% of the aboveground C stock (Progar et al., 2000; Pfeifer et al., 2015; Wu et al., 2020). Deadwood is also considered a stable C pool due to its long residence time (Pfeifer et al., 2015) and provides ecological services such as habitat for plants and soil fauna (Gale, 2000; Woldendorp and Keenan, 2005; Yan et al., 2006; Liu et al., 2006; Gómez-Brandón et al.,
2017; Kumar et al., 2017). Despite its global importance, deadwood remains an understudied terrestrial carbon pool (Gale, 2000; Pfeifer et al., 2015).

Tropical deadwood is mainly cycled biotically through activities of wood-dwelling microorganisms, such as fungi, and invertebrates such as termites (Ulyshen, 2016; Griffiths et al., 2019; Zanne et al., 2022). Invertebrates are responsible for the mechanical breakdown of wood, while fungi and other microbes secrete digestive enzymes to break down wood chemically
(Ulyshen, 2016). The activities of these decomposers are controlled by site-specific environmental conditions (Zhou et al., 2007). Moisture and temperature affect microbial (Hu et al., 2017) and termite activity (Cheesman et al., 2018; Clement et al., 2021; Kim et al., 2021; Zanne et al., 2022) as well as fungal species composition and richness (Pouska et al., 2017; Olou et al., 2019; Dossa et al., 2021), by modulating enzyme production and activity (Pichler et al., 2012; Green et al., 2022) and defining microhabitats suitable for microbial and invertebrate activity (Yoon et al., 2015). Thus, these two variables indirectly
affect deadwood degradation by modifying degradation rates (Hagemann et al., 2010; Hu et al., 2018). Quantifying how environmental conditions influence deadwood degradation rates is necessary to understand the variation of $CO_2$ fluxes from tropical forests across time and space (Cornwell et al., 2009).

There is little consensus around which factors control deadwood degradation and $CO_2$ fluxes from decaying deadwood. Chambers et al. (2000) found that temperature is the best predictor of $CO_2$ fluxes from decaying wood in forests. However,
according to Rowland et al. (2013), this might only be true for temperate forests where stronger temperature gradients are observed, whereas moisture levels are more consistent. The interaction of these two factors could also be important in controlling deadwood degradation rates (Forrester et al., 2012). Precipitation, coupled with high moisture content, increases degradation rates only at high temperatures (Seibold et al., 2021), and high temperatures compensate for slower degradation rates under dry conditions by increasing enzyme kinetics (A'Bear et al., 2014).
Most studies use climate variables, such as air temperature and precipitation, to represent the microclimate where deadwood decay occurs and predict $CO_2$ fluxes from decaying wood (Chambers et al., 2000; Zhou et al., 2007; Hu et al., 2018; Cheesman et al., 2018; Kim et al., 2021). Even though there is a clear coupling between climate variables and microclimate, unique





microclimate conditions may occur under the forest canopy (Floriancic et al., 2023). Few studies have measured wood moisture content and temperature directly, and those that have are limited to a low temporal resolution or impacted by wood degradation

processes if using sensors (Woodall et al., 2020; Green et al., 2022). A low temporal resolution of microclimate variables might mask daily and seasonal variations of wood moisture content and temperature. Consequently, variations of $CO_2$ fluxes from deadwood decay will not be well represented (Green et al., 2022), impeding our understanding of the C budget from forest ecosystems.

In this study, we predict $CO_2$ fluxes from tropical deadwood degradation using wood microclimate variables, which we

define in this paper as wood moisture content and temperature. Taking advantage of gridded climate data from remote sensing (Stackhouse, 2006; Nguyen et al., 2019) and mechanistic fuel moisture content (FMC) models, typically used for firefighting and forestry management (Matthews, 2014), we simulate wood microclimate across a precipitation gradient in Australia. These models use weather variables (air temperature, rainfall, solar radiation, air humidity, and wind speed) to estimate FMC and temperature (Nelson, 2000; Matthews, 2006), explaining up to 94% of the variance in measured FMC (van der Kamp et al.,

2017). Considering the importance of moisture in deadwood decay and given the availability of FMC models to predict FMC from climate, FMC models are a good candidate for downscaling weather variables to wood microclimate for predictions of $CO_2$ fluxes from deadwood decomposition (Figure 1).

To evaluate the link between weather data, wood microclimate, and $CO_2$ fluxes from deadwood decomposition, we adapted a mechanistic FMC model by van der Kamp et al. (2017) to simulate wood microclimate variables, wood moisture content and

temperature along a precipitation gradient spanning dry savanna to wet rainforest ecosystems. In this paper, we will refer to deadwood moisture data collected through processing experimental wood blocks as "moisture content" and data collected by the Campbell CS506 moisture stick as "FMC" data. We hypothesize wood microclimate variables to positively correlate with $CO_2$ flux and cumulative mass loss of pine blocks to correspond to cumulative $CO_2$ flux predicted from wood microclimate variables. Alternatively, if other pathways of mass loss are active, such as termite decay, then cumulative mass loss should

exceed predictions of cumulative $CO_2$ flux from deadwood. Finally, we provide additional mechanistic explanations of factors influencing deadwood decomposition in our study site.



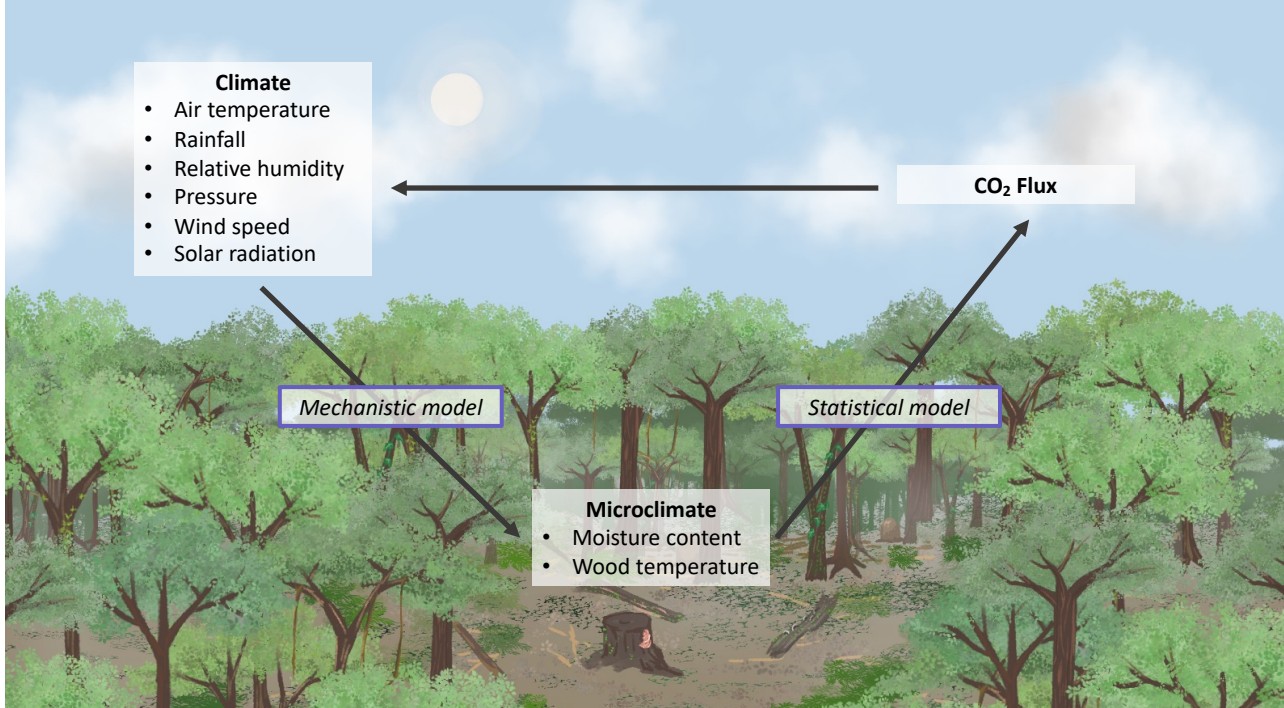

**Figure 1.** Conceptual model of interactions between wood microclimate (defined as wood moisture content and temperature) and $CO_2$ fluxes from decaying wood. Weather variables influence wood microclimate variables, which in turn influences deadwood degradation and the release of $CO_2$ back to the atmosphere. Finally, altered $CO_2$ concentration in the atmosphere affects local and regional climate, influencing future climate patterns. In this study, we used a mechanistic model to derive wood microclimate variables from weather data and a statistical model to relate wood microclimate variables to $CO_2$ flux.

## 2 Methods

### 2.1 Study Site and experimental design

#### 2.1.1 Site description

The study was conducted at five sites along a 75 km rainfall gradient (960-4250 $\mathrm{mm\,year^{-1}}$) in tropical Northeast Australia from June 2018 to June 2022 (https://www.bom.gov.au, 1989-2019). From greatest to least rainfall, the sites (Figure 2) are: James Cook University's Daintree Rainforest Observatory (wet rainforest; 16.1012°S, 145.4444°E) and Australian Wildlife Conservatory's Brooklyn Sanctuary's Mt. Lewis Rainforest (dry rainforest; 16.5933°S, 145.2743°E), Mt. Lewis Sclerophyll (sclerophyll; 16.5830°S, 145.2620°E), Station Creek (wet savanna; 16.610°S, 145.2400°E), and Pennyweight Outstation (dry 85 savanna; 16.5746°S, 144.9163°E). Site descriptors (e.g. wet, dry) are relative to our gradient.





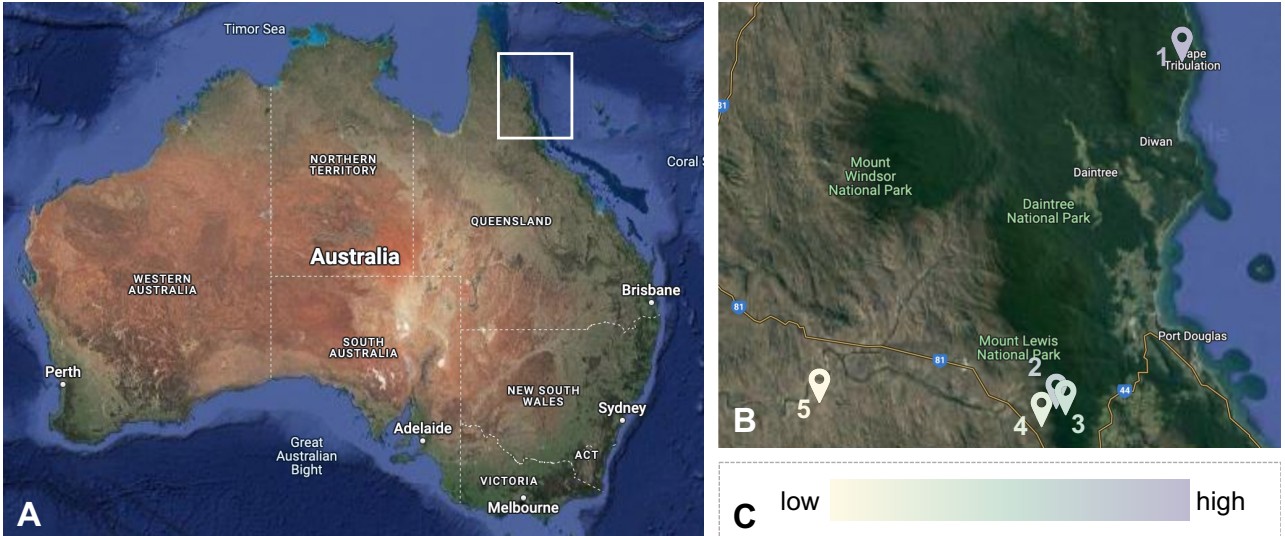

**Figure 2.** Study sites across the precipitation gradient in Australia. The five study sites are in northeastern Australia (Panel A) along a 75 km rainfall gradient. In panel B, sites are numbered from wettest to driest (1. wet rainforest, 2. dry rainforest, 3. sclerophyll, 4. wet savanna, 5. dry savanna). Images are obtained from © Google Earth (https://earth.google.com/). (Panel C) Color schemes used to differentiate sites. Lower precipitation is indicated by lighter color.

### 2.1.2 Pine and common garden experimental setups

Two experiments were started in June 2018: a pine experiment and native species common garden experiment. The pine experiment was set up to determine if $CO_2$ fluxes from coarse woody debris differed across all 5 sites in the rainfall gradient. At each site, pine (*Pinus radiata*) blocks (9 x 9 x 5 cm) were deployed in five plots. The pine block sizes were smaller than the standard definition of coarse woody debris (7.6-10 cm diameter (Harmon and Sexton, 1996; Woodall, 2010; Palace et al., 2012) to facilitate gas flux measurements. For each timepoint, two blocks were deployed in each plot: one enclosed in 280 $\mu$m lumite® mesh (BioQuip) to exclude insect activity and another covered in mesh with 10 holes, 5 mm in diameter, to allow insect access. Blocks were deployed and harvested at 6, 12, 18, 24, 30, 36, 42, and 48 months to capture seasonal variation in $CO_2$ fluxes (2 treatments * 8 harvests * 5 plots * 5 sites = 400 blocks deployed).

The native species common garden experiment included a similar experimental setup to the pine experiment, with wood stems only deployed in the driest and wettest site. Stems ($\sim 7$ cm diameter and $\sim 10$ cm length) of native trees were used to assess variation in decomposition across sites. Native stems were harvested and placed either in wet rainforest (10 species) or dry savanna (6 species) at the sites where they were harvested. There were no overlapping species between sites. Stems were harvested after 12, 18, 24, 30, 36, and 42 months (Table S3). Additional details of the experimental setup are described in Law et al. (2023).




### 2.1.3   Harvest and $CO_2$ flux measurements

During harvests, blocks and stems were removed from their mesh bags, and any accumulated organic matter was removed. Wood pieces were examined for termite and soil presence. An initial field weight was taken, and $CO_2$ flux from the wood was measured with an infrared gas analyzer (Los Gatos Ultraportable Greenhouse Gas Analyzer with a LI-COR Long Term Chamber, Model 8100-104). We used a soil collar (20 cm diameter) to which we affixed a plexiglass bottom. The bottom was used to create a closed chamber and we ensured there were no leaks. After the wood sample was equilibrated in the chamber for 60 seconds, $CO_2$ concentration (ppm) and chamber temperature (°C) were measured over 180 seconds. Block or stem volume was then measured using water displacement. Each wood sample was separated into intact wood, carton (created from termite activity), soil (any soil that entered the bag), and excess (wood shavings and chips) with each component weighed individually. As we were only interested in $CO_2$ fluxes coming from wood, samples which were majority soil or carton were removed from analysis (Figure S1). Final mass was determined after stems and blocks were dried in an oven at 100°C to a constant weight. The proportion of mass loss was calculated using the following formula:

$$Proportion\ Mass\ Loss = \frac{Initial\ dry\ weight - Harvest\ wood\ dry\ weight}{Initial\ dry\ weight} \tag{1}$$

The $CO_2$ flux rate was calculated using the formula derived by Dossa et al. (2015):

$$R_S = \Delta CO_2 \cdot \frac{P}{P_i} \cdot M \cdot \frac{(V_c - V_s)}{V_i} \cdot \frac{T_i}{(T_i - T_c)} \cdot \frac{1}{W_s} \tag{2}$$

where RS is the respiration rate, $\Delta CO_2$ is the change in the concentration of $CO_2$ over time, $M$ is the molar mass of $CO_2$ (44.01 $g\,mol^{-1}$), $P$ is the pressure, $P_i$ is the standard pressure (1013.25 mbar), $V_c$ is the volume of the chamber (4.27 L or 4.58 L), $V_s$ is the volume of the stem, $V_i$ is the standard volume (22.4 L), $T_i$ is the standard temperature (273.15 K), $T_c$ is the temperature of the chamber in °C, and $W_s$ is the dry weight of the stems or wood blocks. $\Delta CO_2$ was determined by taking the slope of the linear fit to $CO_2$ readings plotted over the first 170 seconds of the 180-second measurement in case of time mismatches between the chamber and software clocks. Samples with insignificant (p>0.05) linear fits were removed from the analysis (Figure S1, 3% of total samples). Additionally, blocks harvested at 6 months were excluded from analysis as block volume was not measured. The final rates were represented in units of $\mu g\,CO_2\,s^{-1}\,g^{-1}$ wood.

## 2.2   Wood microclimate: observations and modeling

### 2.2.1   Wood microclimate observations

We considered the wood microclimate variables, wood moisture content and temperature, during flux measurement. We used $T_c$, the temperature of the LI-COR chamber and ambient temperature during flux measurements, to represent wood temperature. Wood moisture content was calculated with fresh and dry weights of intact wood using the following formula:





$$Moisture\ content = \frac{Fresh\ Weight - Dry\ Weight}{Dry\ Weight} \cdot 100\% \tag{3}$$

### 2.2.2 Wood microclimate modeling

a) Model description:

We modeled wood microclimate variables using the fuel moisture model of van der Kamp et al. (2017). Briefly, the model considers a standard stick for fuel moisture measurements to be divided into an inner core ("c") and an outer layer ("o"). The inner core and the outer layer exchange energy and moisture, but only the outer layer exchanges energy and moisture with the environment.

The main components of the energy budget of the stick are i) the incoming longwave radiation ($L_{abs}$), ii) diffuse ($K_{abs-diff}$) and direct ($K_{abs-dir}$) shortwave radiation, iii) emitted longwave radiation ($L_{emit}$), iv) sensible ($Q_h$) and latent ($Q_e$) heat flux, and v) heat conduction ($C$) to and from the stick core. The main components of the moisture budget of the stick are i) the incoming precipitation ($P_{abs}$), ii) evaporation flux from the stick ($E$), and iii) moisture diffusion ($D$) to and from the stick core. Model outputs include temperature and moisture.

The original model expresses all energy fluxes in $\mathrm{W\,m^{-2}}$ and all moisture fluxes in $\mathrm{kg\,s^{-1}}$. However, because our climate dataset was constructed at an hourly temporal resolution (Duan et al., 2023), we adjusted all model energy and moisture fluxes to be expressed in $\mathrm{J\,m^{-2}\,h^{-1}}$ and $\mathrm{kg\,h^{-1}}$, respectively. Similarly, all model parameters and time-dependent parameters are expressed in units per hour (Table S3).

A detailed description of the model formulations is found in (van der Kamp et al., 2017). We present the following minor modifications:

1) Canopy emissivity ($\varepsilon_c$):

We introduced an empirical approach to simulate canopy emissivity dependent on leaf and soil emissivity as proposed by Francois et al. (1997):

$$\varepsilon_c = (1 - c_v) \cdot \varepsilon_g + c_v \cdot \varepsilon_v \tag{4}$$

where: $\varepsilon_g$ [-] is the ground emissivity fixed to 0.95 (Francois et al., 1997), $\varepsilon_v$ [-] is the vegetation emissivity fixed to 0.965 (Francois et al., 1997), and $c_v$ is the contribution coefficient of the vegetation set to 0.5.

2) Precipitable water content ($w$):

Precipitable water content was determined using Pratra's empirical approximation (Prata, 1996):

$$w = C_e \cdot HP \tag{5}$$



where: $C_e$ is an empirical parameter that, unlike in van der Kamp et al. (2017), was set to $46.5 \, \mathrm{cm \, K \, hPa^{-1}}$ for robust predictions (Prata, 1996), and HP is the humidity parameter $[\mathrm{hPa \, K^{-1}}]$:

$$HP = \frac{e_0}{T_a} \tag{6}$$

where $e_0$ [kPa] is saturation vapor pressure calculated with the Magnus-type equation described in Alduchov and Eskridge (1996) and Koutsoyiannis (2012). $T_a$ (°C) is the ambient temperature.

3) Attenuation of shortwave radiation by the canopy:

We incorporated the effect of the canopy on shortwave solar radiation using the approach of Musselman et al. (2015):

$$K_{abs} = K_{abs-diff} \cdot \tau_{diff} + K_{abs-dir} \cdot \tau_{dir} \tag{7}$$

where $K_{abs}$ is the downwelling shortwave radiation measured at the sub-canopy surface, $\tau_{dir}$ is the canopy transmittance of the direct shortwave component that equals the sky-view factor ($sv$) of the canopy, and $\tau_{diff}$ is the canopy transmittance of the diffuse shortwave component calculated as follows:

$$\tau_{diff} = exp\left(-\frac{\xi \cdot \phi \cdot cos(\phi) \cdot \left(exp\left(-\frac{sv - 0.45}{0.29}\right)\right)}{sin(\phi)}\right) \tag{8}$$

where: $\xi$ is an empirical coefficient for calibrated for pine and set to 1.081 (Pomeroy et al., 2009) and $\phi$ is the solar elevation angle in radians.

The model was written in MATLAB (2019), and the $ode15s$ solver was used to solve the differential equation system of the fuel moisture model.

b) Model calibration – data description:

Fuel moisture sticks (Campbell CS506, 10 h Fuel Moisture Stick) were placed at each site to measure fuel moisture content. To simulate the conditions blocks were experiencing, the sticks were placed in mesh bags directly on the ground and annually replaced to avoid measurement errors due to decomposition of the sticks. Due to the fuel moisture sticks' direct contact with the ground, we regularly recorded FMC values above the normal operating range (0-70%). We accounted for this discrepancy by normalizing the recorded values to the operating range.

c) Model calibration – calibration process:

We performed a site-specific calibration for all sites except for the wet rainforest site, using hourly time-series of FMC (see Model calibration-data description). Observations from 2019 were used for calibration, and the remaining data were





used for visual validation of the model results. The wet rainforest site was excluded from calibration due to malfunction of the fuel moisture sensor. Instead, calibrated parameters from the dry rainforest site were used to simulate fuel moisture in the wet rainforest site.

We calibrated five model parameters as in van der Kamp et al. (2017) (Table 1). Fixed parameters (Table S3) and initial conditions of the state variables were taken from van der Kamp et al. (2017) and were set equal for all the sites. Forcing variables (see weather data section) were derived from weather data following the equations suggested by van der Kamp et al. (2017). Parameter ranges were initially taken from van der Kamp et al. (2017), but we extended the parameter ranges to account for the variations of the parameters due to local conditions.

**Table 1.** Model parameters for calibration

| Model parameters | Description | Units | Min | Max |
|---|---|---|---|---|
| $A$ | Empirical constant | [-] | -8 | 20 |
| $B$ | Empirical constant | [-] | -50 | 5 |
| $d_s^*$ | Bulk diffusion coefficient of the stick | $m^2\,d^{-1}$ | $1.0 \cdot 10^{-7}$ | $1.0 \cdot 10^{-4}$ |
| $m_{max}$ | Maximum moisture content of the stick | [-] | 0.1 | 2.5 |
| $f$ | Fraction between core and outer layer of the stick | [-] | 0.05 | 0.95 |

\* $d_s$ was expressed in log scale to facilitate model calibration. Ranges of the parameters were adjusted from van der Kamp et al. (2017).

We used the nonlinear optimization algorithm in MATLAB (2019) $fmincon$ to find the best possible fits of the parameters (Table 1) and the sum of squared errors ($SSE$) as the objective function to compare model output with observations. We estimated the observations' errors from the fuel moisture accuracy of the CS506 Fuel Moisture sensor manual:

$$SEE = \sqrt{\sum_{i=1}^{n} \frac{(simulation_i - observation_i)^2}{error_i}} \qquad (9)$$

The model output corresponding to the observations was the average moisture of the stick $m_s$ (unitless), calculated from
the simulated moisture content in the core and the outer layer of the stick converted to a fraction of the dry weight of the stick:

$$m_s = \frac{(f \cdot m_o + (1 - f) \cdot m_c) \cdot 100}{\rho_s \cdot V_s} \qquad (10)$$

where: $m_c$ [kg] is the moisture content of the core, $m_o$ [kg] is the moisture content of the outer layer, $f$ (unitless) is the fraction of volume between the core and outer layer of the stick, $\rho_s$ is the stick density fixed to $400\,kg\,m^{-3}$ (Nelson,
2000), and $V_s$ [cm$^3$] is the volume of the stick.

d) Wood microclimate simulations:



We simulated pine block moisture content and temperature using the described mechanistic model and the fitted model parameters for each site. Moisture content observations that were measured for pine blocks throughout the experiment were used as the benchmark reference for model performance. Chamber temperature, the closest variable to wood tem-
perature, was used to benchmark wood temperature simulations. There was no additional automatic model calibration of the fuel moisture mechanistic model previously described, only minor modifications to capture the benchmark observations. First, the original geometry of the wood blocks was assumed to be cylindrical with dimensions of 7 cm in diameter and 10 cm in length. Additionally, the parameter $m_{max}$ was manually increased based on field FMC measurements to allow the blocks to hold more water, and the parameter $f$ was manually reduced to allow a more stable moisture content
compared to the sticks due to the higher contribution of the inner core to the final simulated moisture content. Wood density was set to $480 \, \mathrm{kg \, m^{-3}}$.

## 2.3   Weather data

We previously constructed a time series dataset of weather variables across our 4-year (from June 2018 to June 2022) field experiments (Duan et al., 2023). For this project, we extracted from Duan et al. (2023) soil surface air temperature, precipitation,
air pressure, wind speed, relative humidity, shortwave radiation, longwave radiation, solar elevation, and solar azimuth as forcing variables for simulations of fuel moisture of sticks and pine block temperature and moisture.

     We also collected sky view factor data (Table S2) by taking photos of the sky from 1 m above the ground at each site with a fisheye lens and calculating the sky view factor using image binarization (Honjo et al., 2019).

## 2.4   Statistical analysis (wood microclimate variables vs. CO$_2$ fluxes)

To simulate high-resolution $CO_2$ fluxes for each site, we developed a mixed nonlinear model using $CO_2$ flux as the response variable, wood moisture content, temperature, and moisture-temperature interaction as fixed effects, and site as a random effect. This model performed better compared to simpler models that excluded temperature and moisture-temperature interaction (code available on https://github.com/Zanne-Lab/WTF-Climate-Flux). We used the wood moisture measurements from the pine experiment and the corresponding chamber temperature observations to construct the model. To account for simulation
uncertainty, we used the Bayesian inference package *bmrs* (Bürkner, 2017, 2018, 2021) in R version 4.0.4 (R Core Team, 2021). The sampler used 5000 iterations, a warm-up period of 2500 simulations, and four chains and assumed a beta distribution for the response variable. A total of 10000 post-warmup draws were performed. We assessed convergence of the model parameters using the R diagnostic ($\hat{R} = 1$) and tracer plots (Figure S2). Model predictions were obtained using 2000 draws of the parameter posterior distribution. Bayesian p-value equivalent is calculated with the package *bayestestR* (Makowski et al., 2019a, b).
230       Our statistical model of wood microclimate variables and $CO_2$ fluxes was based on observations of *Pinus radiata* blocks, a nonnative common wood bait. To estimate to what extent our model could capture $CO_2$ fluxes of native species, we plotted flux measurements of native species with our statistical model and simulations. Natives were only deployed at the two extremes of our precipitation gradient (wet rainforest and dry savanna), and no species overlapped between sites. We assessed if the relationship between wood microclimate variables and $CO_2$ fluxes measured in pine blocks captured that of native stems at the



wettest and driest sites. Finally, we assessed if $CO_2$ fluxes from native species could be predicted by this pine-based statistical model by plotting the measured $CO_2$ fluxes and wood microclimate variables of native stems together with the modeled and predicted values.

## 2.5  Estimated wood mass loss

We estimated the cumulative mass loss of our pine blocks and native stems at each biannual harvesting point by integrating
our hourly-predicted $CO_2$ fluxes over time. We used the AUC (area under the curve) function and the trapezoid method implemented in the R package *DescTools* (Signorell et al., 2023). We then converted these values from $\mu g\,CO_2\,g^{-1}$ wood to $g\,C\,g^{-1}\,C$ as follows:

$$\frac{\mu g\,CO_2}{g\,wood} \cdot \frac{1\,g\,CO_2}{1000\,\mu g\,CO_2} \cdot \frac{12.01\,g\,C}{44.01\,g\,CO_2} \cdot \frac{100\,g\,wood}{49.2\,g\,C} = \frac{g\,C}{g\,C} \tag{11}$$

First, $\mu g\,CO_2$ was converted to $g\,C$ using the molecular weights of C and $CO_2$. The carbon percentage of *Pinus radiata*,
49.2%, was used to convert g wood to $g\,C$ (Law et al., 2023). The final unit, $g\,C\,g^{-1}\,C$, is comparable to the proportional mass loss measured in field experiments at each harvest time point (Section 2.1.3).

## 3  Results

### 3.1  Wood microclimate validation

We defined wood microclimate variables as wood moisture content and temperature and derived these variables from a fuel
moisture model (van der Kamp et al., 2017) calibrated with fuel moisture stick measurements along a precipitation gradient in Australia (Figure 3). Throughout the four-year experiment, we observed higher wood moisture content at sites with higher precipitation (Figure 3 A). We obtained wood moisture content and temperature simulations that captured major trends in the empirical measurements. Our wood moisture content simulations were sensitive to rainfall events but did not capture the highest block moisture measurements, especially at the wettest sites (Figure 3 A). Moisture values were calculated relative to
the dry weight of the wood (eq. 3). For this reason, moisture can reach values over 100%.

Wood temperature simulations were benchmarked against air temperature at the soil surface. Simulated wood temperature was higher than air temperature at each site and increased with decreasing precipitation regimes, i.e., at dry and wet savannas (Figure 3 B).




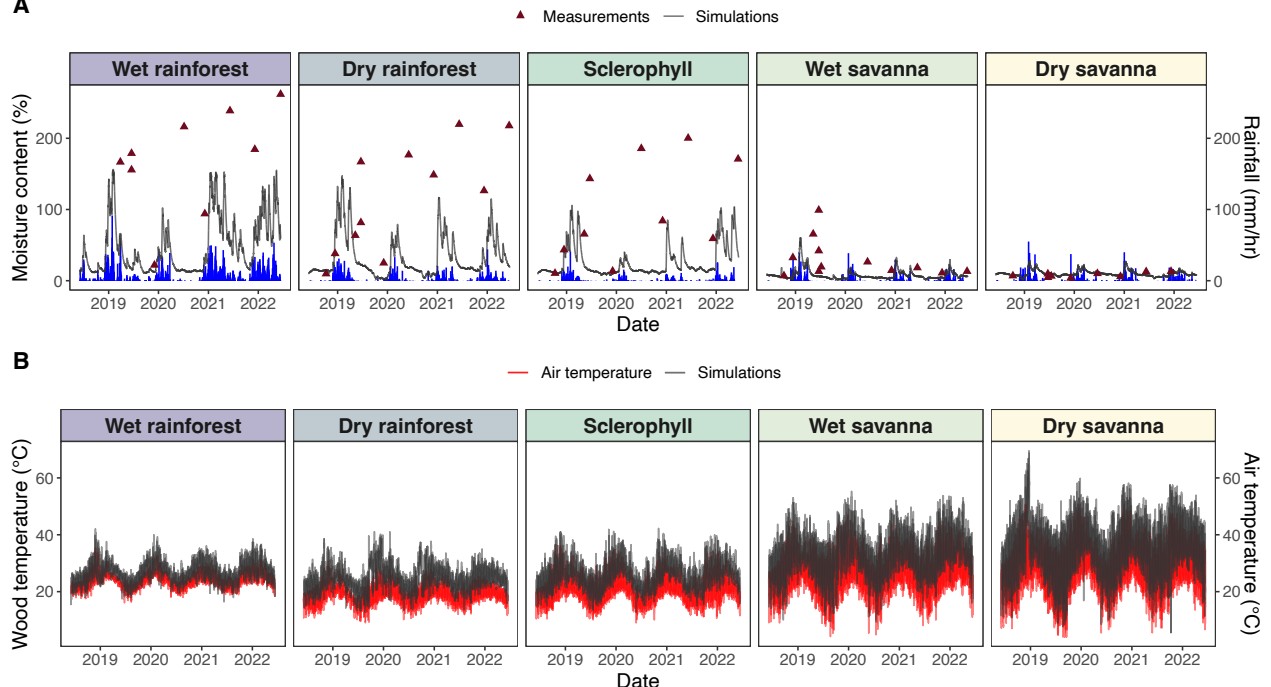

**Figure 3.** Time series of comparisons between pine block simulations and climate observations. (A) Simulated moisture content is shown in gray and hourly precipitation is shown in blue. Different colors represent different sites and triangles represent wood moisture content measurements from field experiments used to calibrate simulations. (B) Simulated wood temperature is shown in gray and soil surface air temperature is shown in red.

### 3.2 Wood microclimate vs. $CO_2$ fluxes across the precipitation gradient

We developed a statistical relationship between wood microclimate variables and $CO_2$ fluxes across our precipitation gradient. Despite the high uncertainty likely attributed to the high variability of the observations (Figure 4 A), our results indicated a positive relationship between $CO_2$ fluxes and wood moisture at each of the study sites (Figure 4 A and Table S1, p-value <0.001). However, the strength of this relationship decreased with decreasing precipitation levels (Table S1). Thus, the savanna sites exhibited lower $CO_2$ fluxes from decaying wood than the rainforest sites. Wood temperature was also positively correlated 265 with $CO_2$ fluxes (Figure S3), but the correlation was not significant (Table S1, p-value = 0.348). On the other hand, the interaction between wood moisture content and temperature was a significant factor in our statistical model (Figure 4 B and Table S1, p-value = 0.001), showing that temperature is relevant, but only when there is sufficient moisture. Therefore, at dry sites, like dry and wet savanna, the temperature is not strongly correlated to $CO_2$ flux due to low moisture levels.





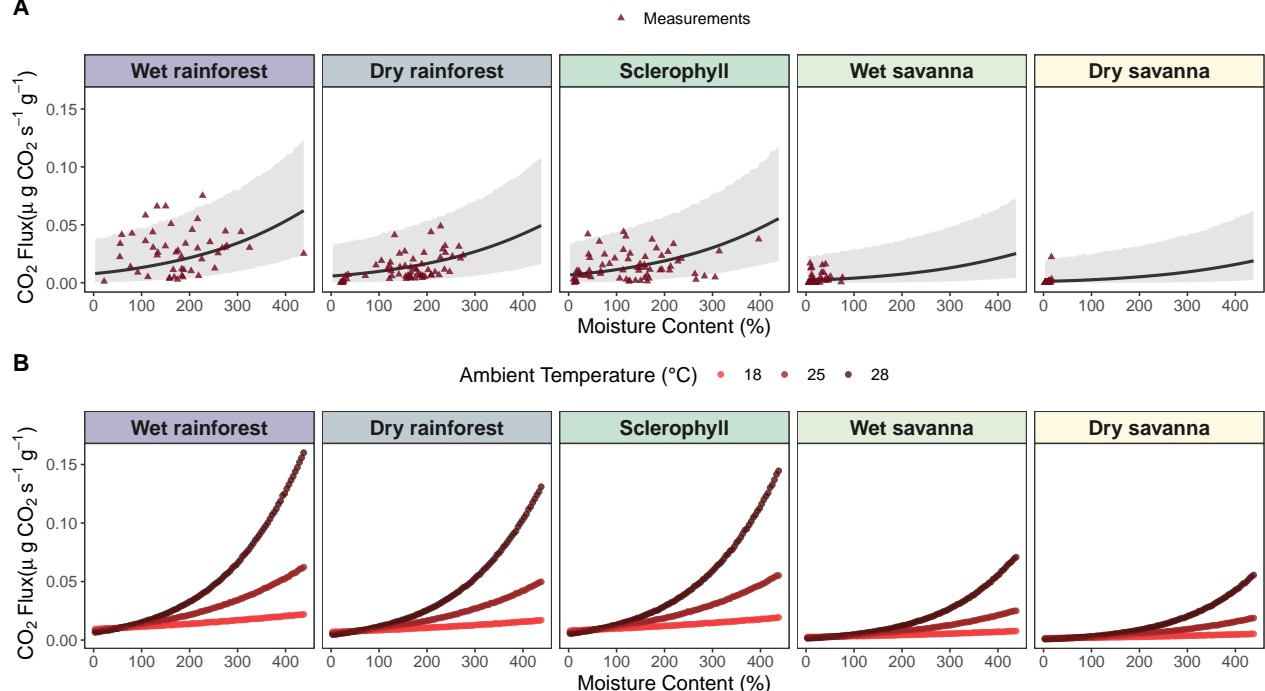

**Figure 4.** Mixed model of $CO_2$ fluxes ($\mu g\,CO_2\,s^{-1}\,g^{-1}$) from decaying wood, with wood moisture content and temperature as fixed effects and site as a random effect. Top row (A): flux predictions against wood moisture content. Bottom row (B): flux predictions against interaction between wood moisture and temperature using three different simulated temperature levels. Triangles represent pine block measurements used to construct the model. An outlier in the dry savanna was kept, as there was no indication that there was an error in measurement.

### 3.3 Time series of $CO_2$ fluxes across the precipitation gradient

We observed patterns in $CO_2$ fluxes, which matched seasonal precipitation patterns (Figure 5). For example, the higher $CO_2$ peaks between 2021 and 2022 in the wet rainforest corresponded to large precipitation events recorded in the area (Figure 3 A). Similarly, in 2022, little precipitation was observed for the dry savanna, which corresponded to lower $CO_2$ fluxes (Figure 5). This seasonal pattern was present at all sites regardless of precipitation regime, although seasonality was more visible at the wetter sites. Wood temperature affected $CO_2$ flux at a daily time scale at all sites, which may have amplified seasonality 275 (Figure 3 B and Figure 5).



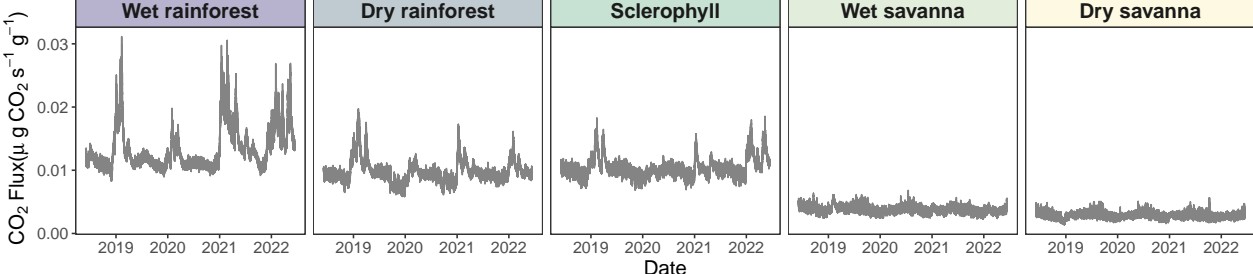

**Figure 5.** High-resolution time series of $CO_2$ fluxes across the precipitation gradient derived from high-resolution time series of simulated wood moisture content and temperature of pine blocks. Solid lines represent model means. Uncertainties were not displayed so seasonal trends would be more apparent (Figure S4).

## 3.4  Simulated and measured wood microclimate: pines vs. native species across the precipitation gradient

Generally, we observed that most native species exhibited a positive relationship between the wood microclimate variables and $CO_2$ fluxes (Figure 6 and Figure S5). This relationship is captured by our statistical model, as measured native $CO_2$ fluxes are within the uncertainty regions of the $CO_2$ estimations (Figure 6 A, C). However, our simulations strongly underestimated $CO_2$

fluxes from decaying native trees (Figure 6 B, D). This was mainly observed in the wet rainforest site (Figure 6 B) and driven by limitations in wood moisture content simulation. In the wet rainforest, simulated moisture content reached a maximum of 200%, whereas measured moisture content surpassed 400%. More measurements of native wood species were captured at the dry savanna site, probably because our simulations successfully captured low moisture content values (Figure 6 D). An exception was the species *Melaleuca viridiflora*, whose stem respiration rates were more sensitive to increasing moisture

content than predicted. In temperature-flux comparisons, our simulations captured a wider range of wood temperature values compared to measurements (Figure S5 B, D), likely because measurements were not always taken on site.



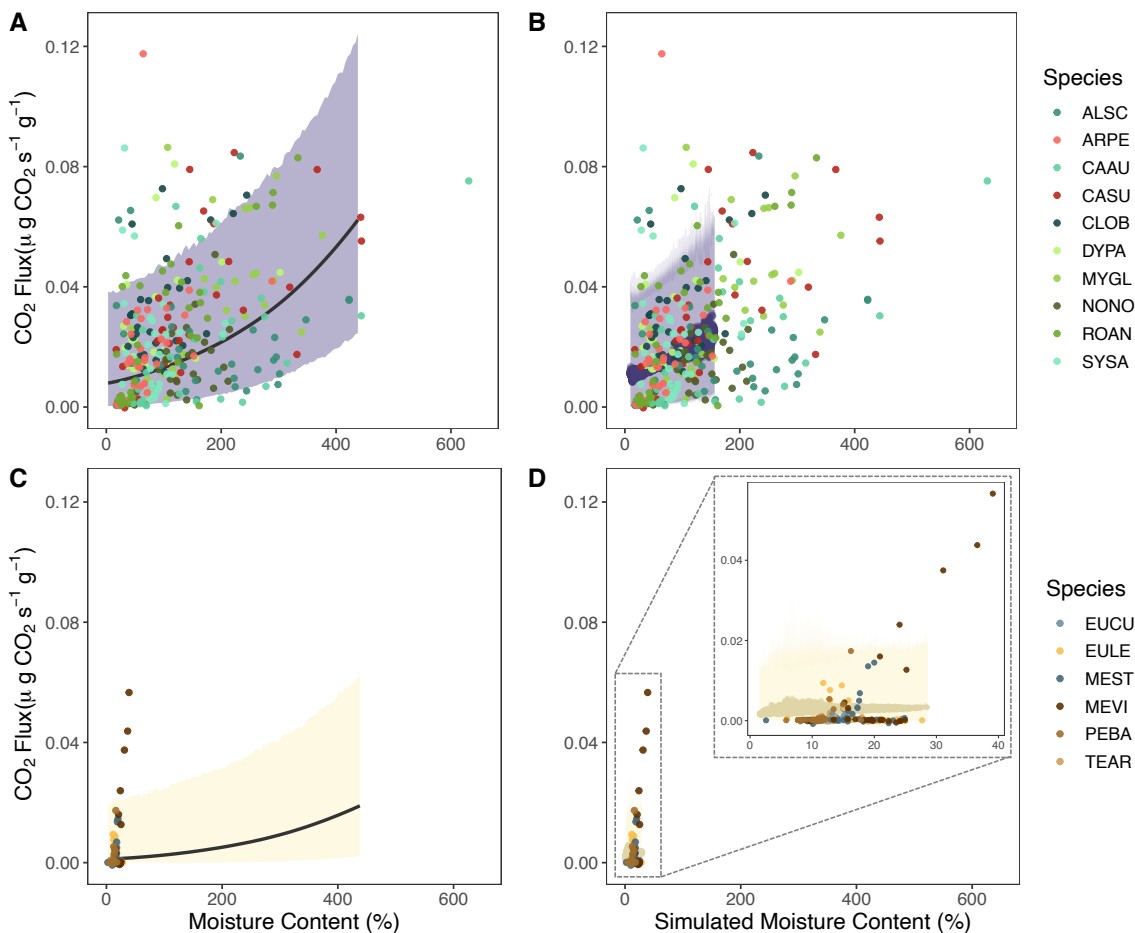

**Figure 6.** Measured native stem moisture content and $CO_2$ fluxes plotted with estimates from the statistical model (A, C) and time-resolved simulations (B, D) Panels A and B show native species found at the wet rainforest panels C and D from the dry savanna. The species name for each code given in Figure 6 is described in Table S4.

### 3.5 Simulated cumulative carbon flux and measured wood mass loss over time

Measured mass loss was positively related to simulated cumulative C flux (Figure 7), with a stronger correlation at wetter sites (R2: 0.92, 0.95, 0.95, 0.60, 0.54 from wettest to driest). We observed a slight overestimation of $CO_2$ flux from decaying wood at the wettest sites. The highest proportion of C released was about 86% of the total block mass in the wet rainforest after 48 months. In the dry savanna at the same time, just 19% of C was released to the atmosphere. If wood blocks that were discovered by termites were included in the analysis, we observed a decrease in the strength of the mass loss-flux correlation (R2: 0.34, 0.86, 0.58, 0.29, 0.41 from wettest to driest) and a clear deviation of the 1:1 line. We additionally ran a linear regression and found a significant interaction between carbon loss and termite activity (p = 0.047, Table S5). This suggests that alternative C





loss pathways besides atmospheric flux directly from decaying wood occurred when termites participate in wood decay. More termite attacks were recorded at the two driest sites (wet and dry savannas), suggesting a higher effect of termite activity at dry sites (Figure 7).

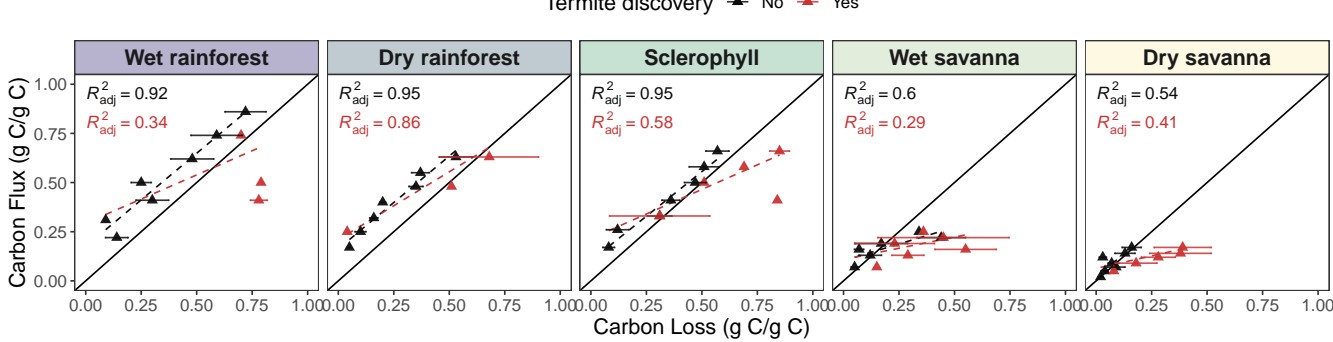

**Figure 7.** Simulated mean cumulative carbon flux [$\mathrm{g\,g^{-1}\,time^{-1}}$] compared with mean measured wood mass loss [$\mathrm{g\,g^{-1}\,time^{-1}}$] of pine blocks. Each point represents a time point in which pine block mass loss was measured (12, 18, 24, 30, 36, 42, and 48 months). The mean carbon loss between blocks at each time point is plotted and bars represent standard error of the mean. Colors indicate whether a termite attack was recorded (red) or not (black). Regression lines and $R^2$ are shown for blocks without termite discovery (black) or all blocks, including those discovered (red).

We observed a similar positive relationship between cumulative $CO_2$ and mass loss for the native woody species, however, cumulative flux did not always equal mass loss (Figure 8). The relationship differed among species, suggesting that wood mass from native species was lost in ways other than as $CO_2$ fluxes, or that our model based on pine is not sufficient to capture the behavior of native species



**Figure 8.** Simulated mean cumulative carbon flux $[\mathrm{g\,g^{-1}\,time^{-1}}]$ and mean measured wood mass loss $[\mathrm{g\,g^{-1}\,time^{-1}}]$ of native species at two ends of the precipitation gradient. Each point represents a time point in which pine block mass loss was measured (12, 18, 24 or 36, and 30 or 42 months, Law et al. (2023)). Points represent mean carbon loss at each time point and bars represent standard error of the mean. Colors indicate whether a termite attack was recorded (red) or not (black). Regression lines and $\mathrm{R}^2$ are shown for blocks without termite discovery (black) or all blocks, including those discovered (red). Species from the Wet rainforest have a blue title background while species from the Dry savanna have a yellow background.



## 4 Discussion

We investigated to what extent wood microclimate variables (wood moisture content and temperature) can describe $CO_2$ fluxes from decaying wood across a precipitation gradient in Australia. First, we obtained high-temporal resolution wood microclimate variables from weather data (air temperature, solar radiation, precipitation, relative humidity, and wind speed) at each site. To use microclimate variables to estimate $CO_2$ flux, we constructed a linear mixed model by correlating wood microclimate variables and $CO_2$ fluxes. Our linear mixed model showed a positive correlation between wood microclimate variables and $CO_2$ that decreased in strength as the weather conditions become drier and hotter (dry savanna). Interestingly, only moisture content and moisture temperature interaction were significant. The positive relationship between wood microclimate variables and $CO_2$ fluxes captured the relationships for most native tree species at the driest and wettest sites, with some exceptions, such as *Melaleuca viridiflora*. Finally, we estimated cumulative carbon flux and compared it to measured mass loss. Consistent with our hypothesis, we observed a positive relationship between mass loss and flux, with termite activity decreasing the amount of carbon released to the atmosphere. Additionally, we found longer wood residence times at drier sites. At the dry savanna, only about 19% of the wood mass loss was released to $CO_2$ within 48 months, compared to 86% at the wet rainforest. Our simple model based on pine species only captured the general patterns of mass loss vs. $CO_2$ exhibited by native tree species. For improved results, species-specific response curves might be required.

### 4.1 Climate variables as predictors of wood microclimate

We estimated wood microclimate variables using a mechanistic fuel moisture model (van der Kamp et al., 2017) driven by weather data measured with portable weather stations or retrieved from gridded databases. Our simulated microclimate variables reproduced the major patterns of the empirical observations (Figure 3), especially the near-ground air temperature patterns. Our model, however, did not capture the measured moisture content peaks. The simulated moisture content reached a maximum of 150%, whereas the measured moisture content, especially at the wetter sites, reached values over 200% and up to 600% in native stems (Figure 6). Wood moisture content was likely sensitive to other physical processes that are not included in the model. This was more evident at the wetter sites, as simulations at drier sites (wet and dry savanna) were closer to the empirical observations. In our case, the wetter sites corresponded to forest ecosystems where microclimates formed under the canopy, which could reduce temperature and evaporation, allowing moisture content to reach high values (Floriancic et al., 2023). Surface runoff, in combination with the topography of the site, could also increase the likelihood of high wood moisture content (Shorohova and Kapitsa, 2016); such topography is present at our wet sites. Additionally, the moisture retention capacity of wood differs among stages of decay (Pichler et al., 2012). This would have required an additional degradation term in the mechanistic model, which is not typical for fuel moisture models and would have added more uncertainty to our simulations.

Nevertheless, our simulated wood microclimate variables were still robust. For example, our simulations showed higher temperatures in wood compared to soil surface air temperature, increasing especially in the hotter and drier sites (Figure 4). This result is consistent with wood thermodynamics by which wood is heated by incoming radiation during the day, and heat is stored and slowly released during the night. Our wood moisture simulations closely resembled the measured moisture content





below 50%, similar to Green et al. (2022). Accurate predictions at these low moisture levels are biologically relevant due to their role in limiting wood decomposition. Excessively high moisture content in wood was not captured by our simulations. We decided not to include additional model terms, such as soil moisture, also because we did not have a complete dataset at our desired temporal and spatial resolution.

## 4.2 Climate-derived wood microclimate as a predictor of $CO_2$ fluxes from decaying wood

We found a positive relationship between wood microclimate variables and $CO_2$ fluxes across the precipitation gradient, and the strength of this relationship decreased at low precipitation sites. This result was expected as wood microclimate variables are known to be important drivers of deadwood degradation (Mackensen et al., 2003; Kahl et al., 2015; Wang et al., 2023), influencing microbial and invertebrate-driven degradation processes (Progar et al., 2000; Zanne et al., 2022). Moisture and temperature modulate enzyme production and activity and determine microhabitats suitable for microbial and invertebrate
activity (Yoon et al., 2015).

As observed in other tropical ecosystems (Wang et al., 2023), we found that wood moisture was an important limiting factor of deadwood degradation at our sites (Table S1, p-value <0.001). Wood moisture controls saprophytic microbial activity (Cheesman et al., 2018) and determines the dominant fungi in decaying wood (Progar et al., 2000; Barker, 2008; Thybring et al., 2018). Bond-Lamberty et al. (2002) found a similar strong correlation between wood moisture and $CO_2$ respiration fluxes but
only below a moisture content of 43%. Similarly, we observed increasing uncertainty in our $CO_2$ predictions with increasing wood moisture content. As wood moisture content increases, the relative importance of wood moisture decreases, and other factors, such as wood traits (wood quality, chemical composition, and stoichiometry), become more relevant (González et al., 2008; Risch et al., 2022; Law et al., 2023). Additionally, high wood moisture contents close to saturation can slow wood decay rates due to anaerobic processes becoming dominant (Piaszczyk et al., 2022).

Chambers et al. (2000) suggested temperature is a better predictor of $CO_2$ fluxes in temperate forests than moisture, arguing that sufficient moisture must be available for trees to grow in the first place. However, in contrast to temperate forests, where wood degradation is limited by temperature, our tropical sites (from wet rainforest to dry savanna) experience relatively similar mean temperatures throughout the year (Figure 3 B), but are subject to very different moisture conditions (Figure 3 A). For this reason, moisture is a limiting factor across our sites and thus is the best predictor for $CO_2$ fluxes. Similarly, Rowland et al.
(2013) found that moisture is the limiting factor for deadwood decay in tropical and subtropical forests. However, temperature variation can interact with moisture and cause $CO_2$ fluxes to be non-linear (Wang et al., 2002; González et al., 2008; Forrester et al., 2012). This is consistent with what we observed: the interaction between wood moisture content and temperature was significant at all sites (Table S1, p-value <0.001) and the relative role of temperature in wood decay increases after a certain moisture threshold is reached (Figure 4 B).

## 365 4.3 Deadwood fate under a precipitation gradient in Australia

An essential question in tropical forest ecosystems is whether the mass loss of woody debris is released to the atmosphere as $CO_2$ or stored in microbial/invertebrate biomass or some other stable form of C (Cornwell et al., 2009). We answered




this question by combining our linear mixed model and high-temporal-resolution simulations of wood microclimate variables. Despite high uncertainty at any given time, when summing $CO_2$ flux estimates over long periods of time, the fine-scale variation

averages out, and estimated cumulative flux was comparable to mass loss of pine blocks (Figure 7). We observed that woody debris has longer residence times in dry, hot sites (wet and dry savanna), and wood decay is enhanced by moisture in wet sites (wet and dry rainforest). Up to 86% of the deadwood is degraded and released as $CO_2$ in the wet rainforest, but less than 19% was released in the dry savanna (Figure 7).

Our model predictions based on wood microclimate variables do not capture invertebrate activity influencing deadwood

decay. When termites are involved in the decay of pine blocks, termite activity lead to deviations from a 1:1 relationship between cumulative $CO_2$ flux and wood mass loss. This suggests that C is lost through other processes which might include leaching, volatilization (Read et al., 2022), and fragmentation (Yoon et al., 2015). These processes may eventually release carbon at locations other than the wood block, for example, termite mounds (Jamali et al., 2013; Clement et al., 2021).

The underprediction of cumulative $CO_2$ flux per mass loss observed in the native stems (Figure 8) suggests that other

biotic factors may need to be included in statistical models when extrapolating beyond wood used for calibration (here, pine) (Jomura et al., 2008). The strength of wood moisture content and temperature influence is likely to vary among tree species (Herrmann and Bauhus, 2013; Wu et al., 2021). Wood traits such as wood nutrient content, quality, and woody debris geometry can be important drivers of CWD decomposition (Zhou et al., 2007; Weedon et al., 2009; Hu et al., 2018; Risch et al., 2022; Kipping et al., 2022). They influence the relative contribution of wood-degrading organisms (bacteria, fungi, and invertebrates

such as termites) and $CO_2$ from wood decomposition. Cornwell et al. (2009) and Law et al. (2023) suggest wood traits, are likely the main determinants of deadwood fate in tropical forests. We found a positive relationship between cumulative $CO_2$ and mass loss holds for most of the native species, however, some species release less $CO_2$ per unit mass loss. This result suggested that the interplay between weather, site conditions, biotic interactions, and specific wood traits (wood quality, chemical composition, and stoichiometry) is essential to determine $CO_2$ fluxes from tropical ecosystems (Law et al., 2023).

For example, mass loss in tree species with dense wood are not accounted for in flux predictions (Figure 8: *Eucalyptus cullenii*, *Eucalyptus chlorophylla*, *Terminalia aridicola*). There are also similar discrepancies for tree species with a high syringyl to guaiacyl (S/G) ratio (*Cardwelia sublimis*, *Normanbya normanbyi*) and species with high nitrogen content (*Rockinghamia angustifolia*, *Petalostigma banksii*).

## 5   Conclusions and implications for global carbon cycling

Wood microclimate variables, defined as wood moisture content and temperature, are essential drivers of deadwood degradation in forest ecosystems. We found that wood moisture content and the interaction between wood moisture content and temperature are the main drivers determining the fate of deadwood degradation along a precipitation gradient in Australia. Because of the high variability in ecosystems and climates within this tropical region, it is essential to consider wood microclimate variables to improve $CO_2$ predictions from decaying woody debris. However, wood microclimate variables alone are insufficient to predict

the $CO_2$ fluxes from diverse native woody species. Additionally, wood traits are likely to be important drivers of CWD fate

in tropical forests (Cornwell et al., 2009; Law et al., 2023) and may improve $CO_2$ predictions in tropical forest ecosystems. Factors such as termite, fungal, and bacterial activity, their climate sensitivity (Zanne et al., 2022), as well as wood traits, such as wood quality, chemical composition, and stoichiometry (Law et al., 2023), and their interplay with climate need to be implemented in future ecosystem models to predict more accurately the fate of woody debris in tropical forests and its

contribution to the global carbon cycle (Cornwell et al., 2009).

*Code and data availability.* The codes and data that support the findings of this study can be found in https://github.com/Zanne-Lab/ WTF-Climate-Flux

*Author contributions.* ESD: Conceptualization, Data curation, Formal analysis, Investigacion, Methodology, Software, Visualization, Writing original draft. LCR: Conceptualization, Data curation, Formal analysis, Investigacion, Methodology, Software, Supervision, Validation,

Writing original draft. NHS: Software, Validation, Review and editing of the draft. BW: Investigation, Validation, Review and editing of the draft. HFM: Data curation, Investigation, Methodology, Sofware, Validation, Review and editing of the draft. AWC: Data curation, Investigation, Validation, Review and editing of the draft. LAC: Investigation, Resources, Review and editing of the draft. MJL: Data curation, Investigation, Review and editing of the draft. PE: Funding acquisition, Project administration, Resources, Review and editing of the draft. AEZ: Conceptualization, Funding acquisition, Project administration, Resources, Validation, Review and editing of the draft. SDA: Concep-

tualization, Funding acquisition, Project administration, Resources, Supervision, Validation, Review and editing of the draft. ESD and LCR contributed equally.

*Competing interests.* The authors have no conflicts of interest related to this article.

*Acknowledgements.* This research was funded by the US National Science Foundation, Ecosystem Studies Cluster, under awards DEB-1655759 and DEB-2149151 to A.E.Z. and DEB-1655340 to S.D.A., as well as UK NERC grant NE/K01613X/1 to P.E. We thank the

Australian Wildlife Conservancy and Daintree Rainforest Observatory of James Cook University for access to field sites. This work was conducted on the unceded territory of the Kuku Yalanji, Djabugay and Djungan peoples, who are the Traditional Owners of the land. We also thank Darren Crayn, Rigel Jensen, and Andrew Thompson for help with species identification; Ana Palma, Emma Carmichael, Paula Gavarró, Gabby Hoban, Jessica Braden, Amy Smart, Xine Li, Baoli Fan, Xennephone Hadeen, Iftakharul Alam, Bethanie Hasse, Hannah Smart, Scott Nacko, Chris Siotis, Tom Swan, Bryan Johnstone, Sally Sheldon, Michaela Fitzgerald, Abbey Yatsko, Rebecca Clement, Mark

Rosenfield, and Donna Davis for help with field work, laboratory analyses and data processing; and Michelle Schiffer and the Cornwell and Wright laboratories for help with logistics.





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
