# Peer review of "Climate-based prediction of carbon fluxes from deadwood in Australia"

_EGUsphere, 2023_

## Author Comment (AC1)

*We thank the reviewer for their insightful and helpful feedback. We have addressed the comments and incorporated the suggestions into our manuscript. In the text below, answers to the reviewer´s comments are written in italics, and changes made to the manuscript text are underlined.*

**Reviewer 1 comments (29 Sep 2023):**

The manuscript is very well written, clearly structured, generally very well illustrated, and covers an highly interesting topic. The presented results are original, novel, and based on an innovative approach. However, before publication can be recommended, the following should be addressed:

**Title, abstract, and elsewhere:**
The terminology is somewhat unclear. The term 'microclimate' usually refers to the conditions (T, RH, prec) in the direct and closest environment of a wooden item. MC and T over time are usually named 'material climate', i.e. the conditions inside the material. The hierarchy is global climate – macro climate – meso climate – local climate – micro climate – materials climate. Suggest to adapt the terminology (see also numerous publications on 'decay modelling of timber structures using this terminology in accordance with ISO standards, e.g. ISO 15686 series).

*We understand the reviewer's concern about the term microclimate, but in ecology and biogeosciences, the term materials climate is unfamiliar. We proposed instead to change the title to "Climate-based prediction of carbon fluxes from deadwood in Australia." Additionally, we have replaced microclimate with "wood moisture and temperature" throughout the manuscript to clarify that we are only referring to these two variables.*

**General comment / Introduction:**
During the last approx. 20 years. Research developed parallels in wood material science and forest ecology. Decay models were developed for timber structures in use (above ground and in soil contact) as well as for deadwood and debris. The intro would significantly improve if similarities and differences between the two approaches would be highlighted. E.g. the hypothesis formulated in L 72-74 has been shown to be correct by different studies in the field of wood material science (e.g. lab tests with pine blocks performed at VTT, Finland, and calorimetry measurements at Lund University, Sweden).

*We agree with the reviewer that our manuscript benefits from including some of the literature in wood material sciences. We have added the following sentences relative to decay models to the introduction and reformulated our hypothesis:*

Line 54: Few studies in ecology have measured wood moisture content and temperature directly, and those that have are limited to a low temporal resolution or impacted by wood degradation processes ...

Lines 72-74: Similar climate-based moisture content models have also been developed for timber structure risk assessment and successfully capture daily and seasonal moisture content trends (Hansson et al. 2012). Our approach has the potential to provide wood moisture and temperature at an hourly time resolution.

Line 77-85: From the perspective of wood integrity and durability, extensive literature in wood material sciences suggests a positive correlation between wood decay and wood materials climate (Viitan 1997, Brische and Rapp 2008a). Additionally, wood moisture and/or temperature are better predictors of wood decay than macroclimate (Brischke et al. 2006, Brischke and Rapp 2008a & 2008b, Niekerk et al. 2021). We extended this idea and further hypothesized that cumulative mass loss of pine blocks corresponds to $CO_2$ flux predicted from wood moisture and temperature. If other pathways of mass loss are active, such as termite-mediated decay, then cumulative mass loss should exceed predictions of cumulative $CO_2$ flux from deadwood. Likewise, we hypothesized that the strength of the relationship between wood moisture and temperature and CO2 fluxes should differ across our precipitation gradient**.**

L9 and 89: Has it been Radiata pine sapwood? Please clarify.

We have added the following sentences to clarify this point:

Lines 99-101: Pine blocks were cut from pine planks obtained from a saw mill. They were harvested from trees grown for timber, so the blocks were likely heartwood. We used this method as we followed a standard protocol for assessing termite activity developed by Cheesman et al. (2018).

Line 109-110: Native stems were harvested directly from our field sites and include heartwood and sapwood (additional details in Law et al. 2023).

Figure 2: The colour scale is hard to interpret. Differences in colour between the test sites are hard to distinguish (even for me, and I am not colour blind).

*We agree the figure was difficult to read. We have raised the saturation of the site colors and added outlines to the markers on panel B.*

[Figure]

L 94: Unclear what is meant with 'treatments'. Is it the application of the mesh? Treatment of wood usually refers to a coating or impregnation with repellents and biocides.

*Yes, we use the term "treatment" to refer to the difference in mesh bags to allow or exclude insect activity. We have reworded the description to clarify thi*s:

Lines 103-106: For each timepoint, two blocks enclosed in 280 μm lumite® mesh (BioQuip) were deployed in each plot to represent two insect access treatments: one completely closed to exclude insect activity and another with 10 holes, 5 mm in diameter, in mesh to allow insect access. Blocks were deployed and harvested at 6, 12, 18, 24, 30, 36, 42, and 48 months to capture seasonal variation in $CO_2$ fluxes (2 insect access treatments * 8 harvests * 5 plots * 5 sites = 400 blocks deployed).

L 128: What is meant with 'intact wood'? Is it non-decayed wood? The wood MC will drastically differ between decayed and non-decayed wood – how has this been considered?

*We use the term "intact wood" to refer to wood blocks that still had structural integrity at the time of harvest, as opposed to wood chips or shavings from blocks shredded by decomposer activity. Wood classified as intact showed signs of decomposition (i.e., changes in porosity, fungal growth), but held their shape and were thus able to be measured for volume and gas flux. We added the following descriptions to clarify*:

Lines 122-123: Each wood sample was separated into intact wood (wood pieces with structural integrity), carton (created from termite activity), soil (any soil that entered the bag), and excess (wood shavings and chips) with each component weighed individually.

**Section 2.2.2:**
It stays unclear how the dimension of deadwood components can be considered.
L 183 / 192: Unclear what fuel moisture sensor is referred to. What kind of sensor? Where installed?

*As mentioned in L 174, we used a Campbell CS506 and 10 h Fuel Moisture Stick to measure fuel moisture content over time. To clarify this, we added the full name of the sensor and change some instances of "stick" to "sensor" or "wood dowel" throughout the manuscript. We also elaborate on our model calibration approach.*

Line 190: Fuel moisture sensors (Campbell CS506 Fuel Moisture Sensor, 10 h Fuel Moisture Stick) were placed at each site to measure fuel moisture content.

Lines 197-199: We followed a two-step calibration approach, in which we first fitted moisture content measured from standard fuel moisture sensors and then derived wood moisture and temperature of cylinders of similar dimensions as our blocks as at hourly resolution**.**

Figure 6: Species codes should be explained in the main text, not only in the supplementary material.

*We have specified the species code and name throughout the manuscript.*

**Discussion, L 303 ff:**

The discrepancy between measured MC and predicted FMC is especially prominent at high moisture levels. Isn't it most likely that this is explained by the geometry of the wooden elements. The effect of capillary water uptake must have been a multiple in the small (more or less cuboid wood block) compared to 'normal' deadwood forming long cylinders. Should eventually be included in the discussion.

*We agree with the reviewer that there are many different aspects that could have led to a mismatch between measured and predicted wood moisture content. As indicated in Line 329, we did not want to include extra terms because it would only have increased the uncertainty of our simulations, which were already high. Nevertheless, we found that the reviewer's suggestion is a valid point, so we added it as follows:*

Lines 359-363: Wood moisture content was likely sensitive to other physical processes that are not included in the model. For example, our blocks were placed flat on the ground, which may have resulted in moisture uptake by capillary action, while our mechanistic model simulated a cylindrical log on its side without accounting for this process, which may have led to underestimates of wood moisture in our simulations (van Niekerk et al. 2021, Thybring et al. 2022)**.**

Lines 439-441: For example, mass loss in tree species with dense wood was not fully captured in our flux predictions (Figure 8: *Eucalyptus cullenii (EUCU), Eucalyptus chlorophylla (EULE), Terminalia aridicola (TEAR)*), likely due to the lower capacity of dense structures to hold water (Thybring et al. 2022)**.**

**General comment / Discussion:**
The link of the presented models/ simulations to the physiological needs of the decay organisms involved is somewhat lacking. Numerous decay models have been developed in Europe and Australia (e.g. at CSIRO) to describe the relationship between wood decay, climate, and a couple of other impact factors. How does all this relate to the findings of the recent study?

*We have added citations to wood materials studies suggested by the reviewer and reformulated our conclusion section to include a comparison of our model results with current carbon models and wood material science decay models.*

Lines 377-378: This result is consistent with wood thermodynamics in which wood is heated by incoming radiation during the day, and heat is stored and slowly released during the night. Sites with higher canopy cover experienced smaller temperature ranges, as shade buffers temperature extremes (Brischke and Rapp 2008b)**.**

Lines 386-389: We found a positive relationship between wood moisture and temperature and $CO_2$ fluxes across the precipitation gradient, and the strength of this relationship decreased at low precipitation sites. This result was expected as wood microclimate variables are known to be important drivers of deadwood degradation (Viitanen 1997, Mackensen et al., 2003; Brischke and Rapp 2008a, Kahl et al., 2015; Wang et al., 2023), influencing microbial and invertebrate-driven decay (Progar et al., 2000; Zanne et al., 2022).

Line 409: However, temperature variation can interact with moisture and cause $CO_2$ fluxes to be non-linear (Viitanen 1997**,** Wang et al., 2002; González et al., 2008; Forrester et al., 2012).

Lines 451-459: Ecosystem-scale carbon models like the YASSO model (Likski et al., 2005) and the CLM soil module (Lawrence et al., 2019) have already incorporated deadwood decomposition as a function of microbial activity affected by climate variables but have not yet explored the effects of wood moisture and temperature on microbial processes related to wood decay. More progress has been achieved in the field of wood material sciences, where the positive correlations between wood moisture and temperature and wood decay has been demonstrated (Brischke et al. 2006, Brischke and Rapp 2008a & 2008b, Niekerk et al. 2021). Our work extends these findings by quantifying the strength of the relationship between wood moisture and temperature and $CO_2$ fluxes from deadwood in response to precipitation and microbial and insect activities. Wood moisture and temperature alone are insufficient to predict the $CO_2$ fluxes, especially from diverse native woody species. Wood traits are likely ...

---

## Author Comment (AC2)

*We thank the reviewer for their insightful and helpful feedback. We have addressed the comments and incorporated the suggestions into our manuscript. In the text below, answers to the reviewer´s comments are written in italics, and changes made to the manuscript text are underlined.*

**Reviewer 2 comments (16 Oct 2023)**

Overall, the manuscript is well-written and easy to follow.

However I have a few issues with wood moisture/temperature modelling that I feel should be addressed.

The largest issue here is that, looking at Figure 3, it's not at all clear to me that the model simulates the observed wood moisture and temperature. Providing the data as time-series, as is done in Figure 3, makes it difficult to assess the model skill. Moreover, nowhere in section 3.1 do the authors provide any model skill statistics (e.g., bias, $R^2$, RMSE, etc). Judging from figure 3A, I would guess that these statistics would not be very promising. As well, the comparison of air temperature and wood temperature in Figure 3B is not very illustrative for the purpose of model validation. I would want to see a comparison of simulated wood temperature and measured chamber temperatures, which the authors take as a proxy for wood temperature. For this manuscript to hold together I feel like the authors should show that the wood moisture/temperature model has a reasonable amount of skill.

*We agree with the reviewer that model skill statistics for wood temperature and moisture are missing. We knew beforehand that the model performance was not going to be optimal due to other processes that are not included in the van der Kamp et al. model that might have played a role in increasing wood moisture in our sites, like subsurface water flow (lines 336-338), and the low density of observations. Consequently, some observations were not correlated to precipitation events and were difficult to capture with our model, leading to non-optimal performance metrics. Therefore, the metrics presented should not be judged too strictly. Nevertheless, we have added a supplementary table S6 with the RMSE and bias calculated using the wood simulations and observations in Figure 3.*

**Table S6.** Model skill metrics (RMSE and Bias) for wood material climate simulations (See Figure 3).

|  | RMSE | | Bias | |
| --- | --- | --- | --- | --- |
| Site | Moisture (%) | Temperature (°C) | Moisture (%) | Temperature (°C) |
| Wet rainforest | 148.9 | 3.7 | -135.7 | 2.9 |
| Dry rainforest | 118.1 | 7.8 | -90.1 | 6.6 |
| Sclerophyll | 102.0 | 6.6 | -77.1 | 6.3 |
| Wet savanna | 34.8 | 12.6 | -23.6 | 10.7 |
| Dry savanna | 4.1 | 11.6 | 2.7 | 10.9 |

*We also plotted Figure 3B against measured chamber temperatures.*

[Figure]

Figure 3. Time series of comparisons between pine block simulations and climate observations. (A) Simulated moisture content is shown in gray and hourly precipitation is shown in blue. Different colors represent different sites and triangles represent wood moisture content measurements from field experiments used to calibrate simulations. (B) Simulated wood temperature is shown in gray and soil surface air temperature is shown in red. Triangles represent the temperature of the LI-COR chamber during flux measurements. Model skill metrics (RMSE and Bias) are presented in supplementary Table S6.

*We have added a Figure in the supplementary Figure S6 to show the original calibration results on stick moisture and the residuals (observations-simulations). We reported the RMSE in the supplementary Table S2.*

[Figure]

Figure S6. Original calibration results on sensor dowel moisture per site (A) and residuals (B).

Another issue (that I believe may be a cause of the poor model skill) is the fact that the authors train the wood moisture/temperature model on a standardized fuel stick, but then apply the model to wood blocks with different dimensions. This may be problematic because in van der Kamp et al (2017) when the model was fit to different sized sticks, the optimal parameter set changed, suggesting that there was no one parsimonious model that could be applied to a piece of wood of arbitrary size. Indeed the authors found the need to re-adjust the m_max and f parameters.

*Please see the answer below.*

This brings me to my next major point: Why not attempt to use the van der Kamp model to simulate the observed wood block moisture values directly, and avoid the step of modelling the automatic fuel stick first? It's not clear to me that this approach would result in worse model skill than what's shown in Figure 3.

*We agree with the reviewer that the best option would have been to use a direct calibration on the wood blocks to avoid uncertainties in the simulations due to the two-step calibration process. Unfortunately, the data density of the wood moisture was too low—we only had a few scattered observations (in some locations, only 8 points) throughout four years. These few points would have impeded obtaining plausible simulations that could capture the dynamics of wood moisture and temperature at hourly intervals, as we could do with the sensor data. Additionally, the use of fuel moisture sensors is a standard practice that can be paired with weather stations. This would allow us to derive wood moisture and temperature variables*

*relatively easily from weather data, knowing that the installation and management of wood blocks in the field is challenging. To address the issues pointed out by the reviewer, we augmented section 4.1 as follows:*

Lines 351-355: We followed a two-step calibration approach, in which we first fitted moisture content measured from standard fuel moisture sensors and then derived wood moisture and temperature of cylinders of similar dimensions as our blocks at hourly resolution. Despite the potential uncertainty in the simulations, this calibration approach was chosen due to the low density of wood moisture observations that limited representation of hourly dynamics of wood temperature and moisture in a single simulation.

SECONDARY ISSUES:
One secondary issue that the van der Kamp model assumes that the stick is elevated above the ground. However, in this study the sticks were sitting on the ground. The main issue with this discrepancy is that for the low wind regime of a subcanopy sites, the aerodynamic resistance of an elevated fuel stick is likely going to be less than for a stick sitting on the ground; moisture and heat are more easily transported to and from an elevated stick. However, your model calibration likely implicitly corrects for this by decreasing the internal diffusivity to compensate for the inflated aerodynamic diffusivity. Indeed, your optimized bulk diffusion coefficient is an order of magnitude lower than the values found by van der kamp et al. If you do use the van der Kamp model for simulating wood on the ground, I would hope to see this issue at least mentioned in the manuscript.

*We agree with the reviewer that the placement of the sticks on the ground is not standard practice. The rationale of the placement in our experiment was to resemble degradation conditions of woody debris, but it is clear, as pointed out by the reviewer, that some of the original physics in the van der Kamp model may not hold. We added more explanation in the methods section "Model calibration – calibration process":*

Lines 208-2011: Parameter ranges were initially taken from van der Kamp et al. (2017), but we extended the parameter ranges to account for the fact that sensors were placed directly on the ground rather than raised above the ground, which may alter original physical properties described in van der Kamp et al. (2017), such as aerodynamic resistance.

*And:*

Lines 371-374: Finally, to simulate more closely the conditions experienced by deadwood, sensor dowels were placed on the ground and not above ground as per standard practice. Representing this variation may have influenced energy and moisture transport described in van der Kamp et al. (2017). We allowed our parameters to take on values beyond the range proposed by van der Kamp et al. (2017) during calibration to account for this issue.

Another issue is the fact that the authors normalized the automatic fuel stick data to remain within the range of operating range reported by Campbel Scientific. I have read numerous

articles that use the CS506 sticks, and have worked with these sticks myself, and I've never come across such an approach. Is this something Campbell Sci recommends? Unless this is something Campbell Sci suggests, or if you have a defensible, physical reason for doing this, I would recommend avoiding this step.

*As we regularly recorded values beyond the operating range of the sticks (above 70%), we decided to normalize the values to reduce the weight of potentially error-prone high measurements. However, we agree with the reviewer that normalizing the data to the operating range is not standard practice, so we removed the normalization and recalibrated the sticks using the raw data. We deleted lines 177-178.*

Another issue with using the CS506 sticks that isn't mentioned here is the fact that individual fuel sticks have consistent biases when compared to each other (see section A.2.2 of van der Kamp, D. W. (2017). Spatial patterns of humidity, fuel moisture, and fire danger across a forested landscape. The University of British Columbia). The mean biases between sticks are often on the order of a few % moisture content. Again, the model calibration would probably compensate for this bias as the authors undertake a separate calibration for each stick. However, these models would likely lead to wet or dry biases when applied to the different wood blocks.

*As pointed out earlier, we do not think that model bias is a very meaningful performance metric for our simulations because of the low density of observations and uncertainty on the precise hour the observations were taken. We believe that our calibration approach was able to account for these differences and capture the seasonal trends of wood moisture and temperature that we were aiming to capture.*

I have a few smaller points. Firstly, the use of a sum of squared errors as a model evaluation metric seems odd to me. Why not use something more common, like RMSE, or MSE? Also, the fact that the metric is a sum and not a mean is confusing. Wouldn't the SSE metric therefore be dependent on the number of datapoints available? That seems less than ideal for a model evaluation metric. Also, a sum of errors doesn't really mean anything that intuitively makes sense physically. Finally, equation 9 defines SSE, but why is there an $error_i$ term as a denominator? is the $error_i$ equal to $simulation_i - observation_i$? If so, this just ends up being the square root of the sum of the errors. I've never heard of a "sum of Squared errors" being used as an objective function for model calibration.

*Initially, we intended to minimize the weighted sum of squared errors because we wanted to penalize observations with higher uncertainty. According to the Campbell user guide, higher moisture values also have higher uncertainty; therefore, the error values ($error_i$) were derived from these suggested values (see line 192). The paper had a typo in the equation and the parenthesis in equation 9 should include the $error_i$ term. Despite this explanation, we agree with the reviewer that RMSE is a more common metric and potentially more applicable for our fitting exercise, given that we do not have repetitions or measured standard deviation of the observations and that, therefore, an error term ($error_i$) might be quite arbitrary to use. We have*

*adopted the suggested RMSE as the new objective function and rerun the calibration. We corrected the text and equation as follows*:

Line (191) … and the root mean square error (RMSE) as the objective function to compare model output with observations (eq. 9)**:**

$$RMSE = \sqrt{\sum_{i=1}^{n} \frac{(simulations_i - observation_i)^2}{n}} \dots \text{(eq. 9)}$$

Finally, A brief description of the weather dataset would be helpful. the Duan et al., 2023 reference doesn't seem to contain a very detailed description.

*We agree that the description of the weather dataset could be more detailed. We added in the following clarification and cited the github repository of Duan et al. 2023 that contained a more detailed description of the weather dataset*:

L 215: We previously constructed an hourly time series dataset of weather variables across our 4-year (from June 2018 to June 2022) field experiments using Vaisala Weather Transmitters (WXT530), gap-filled with publicly-available weather datasets (Duan et al., 2023, detailed methods available on https://github.com/Zanne-Lab/WTF-Climate)**.**

---

## Referee Report (RR1)

Thank you for taking the time to address my comments.

I appreciate that you include some model statistics in your revisions. However, what the RMSE and bias values do not show is that, based on my assessment, the fuel stick moisture model does not have any predictive skill when it comes to simulating the observed wood block moisture values.

Using the data and scripts in the github repo, I was able to reproduce Figure 3a. I then calculated the Nash-Sutcliffe efficiency metric for each site individually. The values were all below zero, which indicates that the model does not do any better than a simple mean value. As well, I ran linear regressions between the simulated and observed data for each site individually and found that the p-values were all larger than 10%. Finally, I did a single regression of simulated and observed data across all sites and found that the R^2 value for the simulated moisture is less than when you regress the observed data against the intra-site averages.

This analysis suggests to me that the model has no real skill in modelling the block moisture. So I would suggest that it is inappropriate to use the model output as the basis of the rest of your analysis.

If desired, I can provide the R-code I used to undertake this analysis.

---

## Author Response (AR2)

*We thank the reviewer for their insightful and helpful feedback. We have addressed the comments and incorporated the suggestions into our manuscript. In the text below, answers to the reviewer´s comments are written in italics, and changes made to the manuscript text are underlined.*

**Reviewer 2 comments (16 Feb 2024):**

Thank you for taking the time to address my comments. I appreciate that you include some model statistics in your revisions. However, what the RMSE and bias values do not show is that, based on my assessment, the fuel stick moisture model does not have any predictive skill when it comes to simulating the observed wood block moisture values. Using the data and scripts in the github repo, I was able to reproduce Figure 3a. I then calculated the Nash-Sutcliffe efficiency metric for each site individually. The values were all below zero, which indicates that the model does not do any better than a simple mean value. As well, I ran linear regressions between the simulated and observed data for each site individually and found that the p-values were all larger than 10%. Finally, I did a single regression of simulated and observed data across all sites and found that the R^2 value for the simulated moisture is less than when you regress the observed data against the intra-site averages. This analysis suggests to me that the model has no real skill in modelling the block moisture. So I would suggest that it is inappropriate to use the model output as the basis of the rest of your analysis. If desired, I can provide the R-code I used to undertake this analysis.

*Upon further review of our analyses, we found a typo in our code that caused the rain flux to be too low (see line 174 from forcing_minutes_wood.m file). Additionally, to better represent the moisture content of wood blocks in the field, we changed the geometry from cylinders to blocks, adjusting some of the equations for heat and water transfer applicable to block geometry (see fuel_model_wood.m and forcing_minutes_wood.m files). After correction and recalibration of the models, the wood moisture simulations based on the fuel moisture stick simulations improved both visually (see Figure 3) and quantitatively based on the Nash–Sutcliffe efficiency metric (Table S2) suggested by the reviewer. Four out of five sites performed at least as well as the mean value of the observations, and only one site had a negative Nash-Sutcliffe score. This suggests that the model simulations and further conclusions are robust.*

*We acknowledge that the Nash–Sutcliffe efficiency metric is an interesting way to assess model performance for time series. However, we note that this metric is typically used for assumptions in hydrological problems and not for such a dynamic variable as $CO_2$. Therefore, it may not be the most appropriate metric for our exercise, especially given the sparse number of data points. Nonetheless, we computed the metric, showing that our model simulations have the potential to perform better than the site mean of the observations. More importantly, we would like to emphasize that the benefits of our mechanistic model extend beyond its predictive capacity. It responds to climatic variables, which allows us to understand $CO_2$ temporal and spatial dynamics at an hourly resolution. With sufficient data, our approach can be applied in various scenarios, such as seasonal comparisons across sites—something that the simpler mean value approach cannot provide.*

*Because the new calibration step slightly changed the subsequent analysis, we have updated the figures and the interpretation accordingly. For example, our new results suggested a more rapid degradation of wood in the wet rainforest and for some native species. However, the overall patterns and conclusions remain the same. Please see the manuscript with the highlighted changes.*

---

## Author Response (AR3)

*We thank the reviewer and the editor for their insightful and helpful feedback. We have addressed the comments and incorporated the suggestions into our manuscript. In the text below, answers to the reviewer´s and editor´s comments are written in italics, and changes made to the manuscript text are underlined.*

**Reviewer´s comments (May 15, 2024):**

I would suggest that a table be added to section 3.1 that outlines, for each of the 5 sites, standard model skill metrics for the fuel moisture model, including, RMSE, R^2, bias, as well as the NSE.
It is important that the authors are up front about the model skill, and that the reader is provided with the information required to assess the skill of the model.
This model is used later in the analysis to predict C02 fluxes, so the onus is on the authors to show that the model has predictive skill.

*We have added a Table in section 3.1, including the standard model skill metrics suggested by the reviewer.*

**Table 2.** Standard model skill metrics for the wood moisture model

| Sites | Performance metrics | | | |
| --- | --- | --- | --- | --- |
| | RMSE | $R^2$ | MBE | NSE |
| Wet rainforest | 70.8 | 0.5 | -44.9 | 0.1 |
| Dry rainforest | 71.2 | 0.1 | -11.1 | 0.04 |
| Sclerophyll | 51.8 | 0.4 | 3.2 | 0.4 |
| Wet savanna | 23.2 | 0.3 | -5.9 | 0.2 |
| Dry savanna | 6 | 0.01 | -0.5 | -2.8 |

RMSE = Root mean squared error, $R^2$ = Coefficient of determination, MBE = Mean Bias Error, NSE = Nash-Sutcliffe efficiency.

**Editor´s comments:**

The corrections you made in the coding and the manuscript led to a substantial improvement which was also acknowledged by reviewer #2. I have decided that still some minor revisions are necessary before the mansucript can be accepted. Please follow the suggestion of the reviewer to add a table to section 3.1 that outlines, for each of the 5 sites, standard model skill metrics for the fuel moisture model, including, RMSE, $R^2$, bias, as well as the NSE. I noticed that your discussion starts with a partial recap of results and methods especially in the first paragraph. I suggest that you remove this because it is not necessary and makes the manuscript unnecessarily long. In a discussion you should discuss results, and not repeat results and methods.

*In addition to the table in section 3.1 (see previous comment), we have deleted the introductory paragraph in the discussion section.*